# Prognosis prediction of uterine cervical cancer using changes in the histogram and texture features of apparent diffusion coefficient during definitive chemoradiotherapy

**Akiyo Takada**[1]*, **Hajime Yokota**[2], **Miho Watanabe Nemoto**[2], **Takuro Horikoshi**[1], **Koji Matsumoto**[1], **Yuji Habu**[3], **Hirokazu Usui**[3], **Katsuhiro Nasu**[1], **Makio Shozu**[3], **Takashi Uno**[2]

**1** Department of Radiology, Chiba University Hospital, Chiba, Japan, **2** Department of Diagnostic Radiology and Radiation Oncology, Graduate School of Medicine, Chiba University, Chiba, Japan, **3** Department of Reproductive Medicine, Obstetrics and Gynecology, Graduate School of Medicine, Chiba University, Chiba, Japan

* r10.aa.0826@gmail.com

## Abstract

### Objectives

We investigated prospectively whether, in cervical cancer (CC) treated with concurrent chemoradiotherapy (CCRT), the Apparent diffusion coefficient (ADC) histogram and texture parameters and their change rates during treatment could predict prognosis.

### Methods

Fifty-seven CC patients treated with CCRT at our institution were included. They underwent MRI scans up to four times during the treatment course (1st, before treatment [n = 41], 2nd, at the start of image-guided brachytherapy (IGBT) [n = 41], 3rd, in the middle of IGBT [n = 27], 4th, after treatment [n = 53]). The entire tumor was manually set as the volume of interest (VOI) manually in the axial images of the ADC map by two radiologists. A total of 107 image features (morphology features 14, histogram features 18, texture features 75) were extracted from the VOI. The recurrence prediction values of the features and their change rates were evaluated by Receiver operating characteristics (ROC) analysis. The presence or absence of local and distant recurrence within two years was set as an outcome. The intraclass correlation coefficient (ICC) was also calculated.

### Results

The change rates in kurtosis between the 1st and 3rd, and 1st and 2nd MRIs, and the change rate in grey level co-occurrence matrix_cluster shade between the 2nd and 3rd MRIs showed particularly high predictive powers (area under the ROC curve = 0.785, 0.759, and 0.750, respectively), which exceeded the predictive abilities of the parameters obtained from pre- or post-treatment MRI only. The change rate in kurtosis between the 1st and 2nd MRIs had good reliability (ICC = 0.765).

**Data Availability Statement:** All relevant data are within the manuscript and its Supporting Information files.

**Funding:** This study has received funding by the Japan Society for the Promotion of Science (JSPS) KAKENHI (grant number JP20K16752) awarded to AT.

**Competing interests:** The authors have declared that no competing interests exist.

## Conclusions

The change rate in ADC kurtosis between the 1st and 2nd MRIs was the most reliable parameter, enabling us to predict prognosis early in the treatment course.

## Introduction

Uterine cervical cancer (CC) is among the most commonly diagnosed cancers in women, resulting in over 300,000 deaths annually worldwide [1]. For International Federation of Gynecology and Obstetrics (FIGO) stages 1B to 4A, concurrent chemoradiotherapy (CCRT) is the current standard of treatment. Image-guided brachytherapy (IGBT) planning has become available, and standard approaches have been developed [2]. In recent years, a treatment plan tailored according to the risk of recurrence and expected adverse events, so-called 'precision medicine', has become common. Three-dimensional (3D) IGBT is a new brachytherapy planning method for CCRT that uses 3D images acquired from MRI and CT. Three-dimensional-IGBT has achieved dose-distribution control according to the shape of individual tumors and has the potential to allow dose adjustment according to recurrence risk [3]. Therefore, accurate recurrence risk prediction is essential to performing precision medicine with 3D-IGBT for CC.

Prognostic factors for CC have included age, race, FIGO stage, histological type and grade, tumor volume, lymph node involvement and location, lymphovascular space invasion, and performance status [1], but their accuracy is limited. Some previous studies found no significant difference in FIGO stage (FIGO2008) and tumor volume between recurrent and non-recurrent disease [4,5]. Three-dimensional-IGBT improves prognosis for advanced bulky tumors [6,7], which may in turn decrease the value of "tumor volume" as a prognostic factor. Because lymph node involvement and lymphovascular space invasion are information that can only be obtained by surgical pathology, which is impossible for patients receiving CCRT, prognostication using images has attracted attention. In particular, the utility of MRI has been reported [8–10].

Several studies have reported the accuracy of apparent diffusion coefficient (ADC) values of the tumor in predicting lymph node metastasis. These values are also helpful in identifying an early response to CCRT, and in predicting mid- to long-term prognosis in CC [9,11,12]. Given that most of these studies used pretreatment MRI scans [9,12] we hypothesized that the ADC histogram and texture features from MRIs during the treatment course and their change rates would reflect the response to treatment and might improve the prognostic accuracy. The ability to determine prognosis early in the clinical course would help in further treatment planning.

The purpose of this study was to investigate whether the histogram and texture features of ADC and their change rates during CCRT could be used to predict the disease course of CC in patients treated with CCRT.

## Materials and methods

### 1. Patients

**1A. Enrolment.** The research ethical committee of Chiba University Hospital approved this study (approval number 2498), and written informed consent was obtained from all patients. We prospectively assessed 125 CC patients who underwent definitive CCRT between

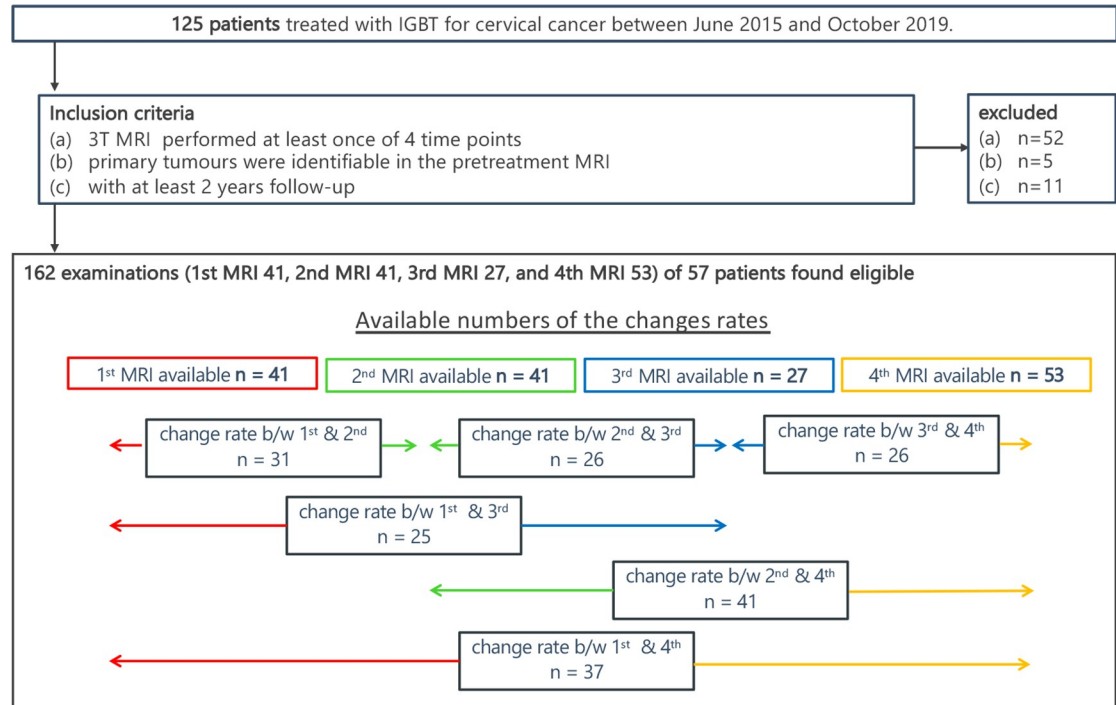

**Fig 1. Flowchart showing patient enrolment and inclusion criteria.** The numbers of MR examinations available at each time point and number of calculated change rates are shown.

June 2015 and October 2019 at our institution (Fig 1). The patients were slated to undergo four MRI scans: 1st MRI—before starting treatment (chemotherapy and external beam radiotherapy (EBRT)); 2nd MRI—just before the start of IGBT (during chemotherapy and EBRT); 3rd MRI—in the middle of IGBT (at the end of EBRT); and 4th MRI—2–3 months after the end of treatment (Fig 2). These time points were determined in consultation with radiation oncologists as important for observing the therapeutic effect. The following inclusion criteria were applied: (a) patients who underwent a 3T MRI (Ingenia, Philips Medical Systems, Best, Netherland) at least one of four time points (b) primary tumor was identified in the pretreatment MRI; and (c) patients with ≧2 years of follow-up. We chose patients with ≧2 years of follow-up because a large proportion (62%–89%) of cervical cancer recurrences were detected within 2 years of primary treatment [13]. A total of 57 patients matched the criteria (Fig 1).

**1B. Treatment.** Treatment strategy was determined in consultation with gynecologists and radiation oncologists according to each patient's clinical status, clinical stage, and histological type. Our institute's standard treatment protocol for CC is the same as those for squamous cell carcinoma (SqCC), adenocarcinoma, and adenosquamous carcinoma. The primary tumor and regional lymph nodes at risk are treated with EBRT at a dose of 50 Gy (2.0 Gy per fraction) combined with concurrent daily cisplatin chemotherapy [14]. External beam radiotherapy was delivered in three-dimensional conformal radiation therapy with an Ir-192 source. The high-risk clinical target volume (HR-CTV) [15], determined after EBRT of 30 Gy [16] was then treated with 18–24 Gy 3D-IGBT (Fig 2). Each brachytherapy session was planned based on CT or MRI obtained at each BT.

**1C. Follow-up.** The patients were followed up by gynecologists and radiation oncologists. In the first two years, pelvic examinations were performed, and cytology and tumor markers (CEA, CA19-9, CA125, SCC antigen) were acquired every 2–3 months. Follow-up CT was

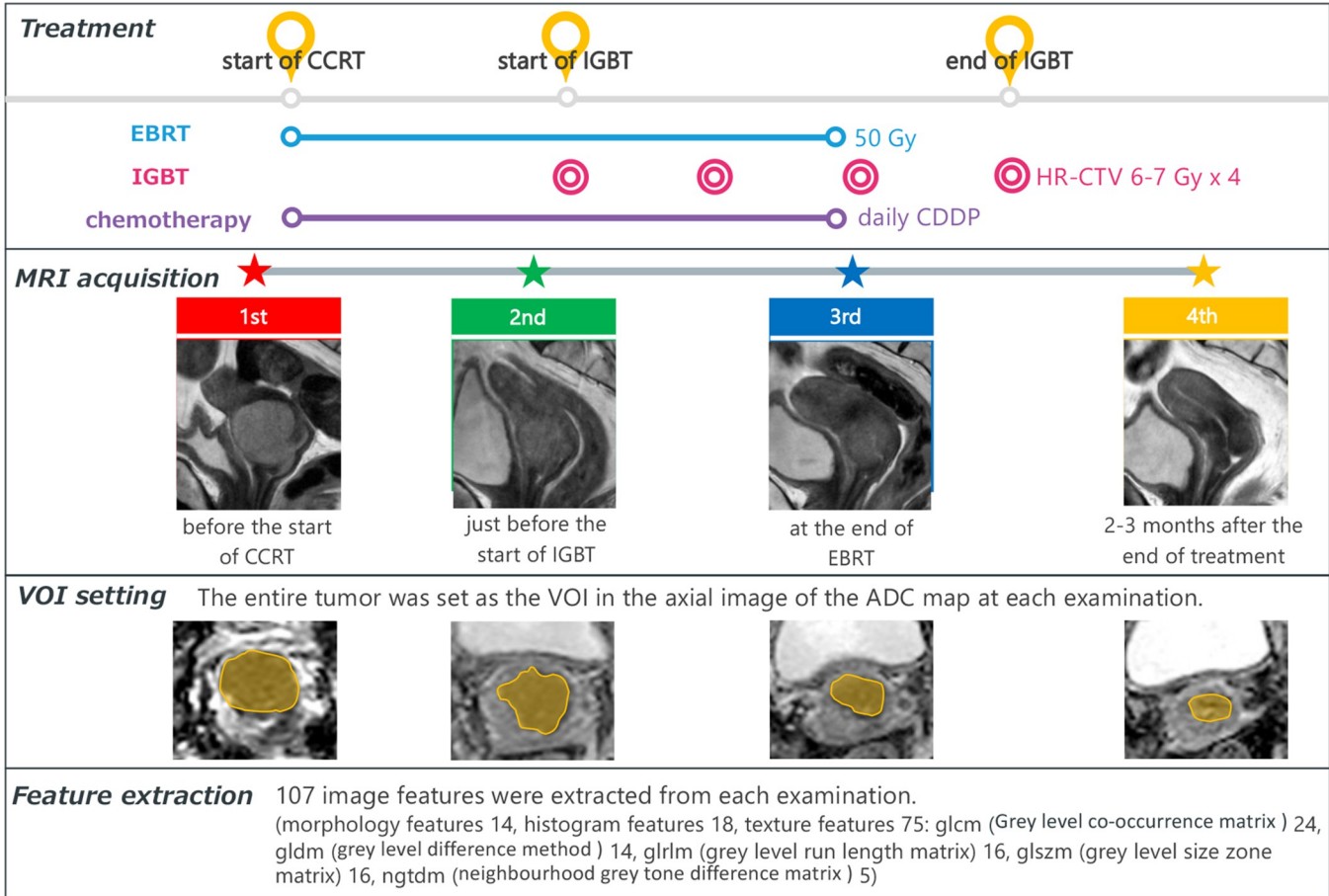

**Fig 2. Flowchart of treatment, MRI acquisition, and imaging processing.** CCRT = cervical cancer radiation therapy; IGBT = image-guided brachytherapy; EBRT = external beam radiotherapy; HR-CTV = high-risk clinical target volume; CDDP = cisplatin.

performed three months after treatment completion and annually for two years. When recurrence was suspected with the above examinations, additional CT, MRI (not counted as a time point), or PET was performed. Recurrences were diagnosed comprehensively based on physical examination, imaging, and clinical course after discussion among gynecologists, radiologists, and pathologists.

## 2. Imaging

**2A. MRI acquisition.** The standardized image acquisition protocol included T2-weighted images (T2WI) T1-weighted images (T1WI), and contrast-enhanced T1WI in the axial and sagittal planes. Diffusion-weighted images (DWI) were acquired in the axial plane. The main scan parameters are shown in Table 1. All MRIs were acquired without brachytherapy applicators.

**2B. Segmentation.** The entire tumor was set as the volume of interest (VOI) in the axial images of the ADC map by a radiologist (10 years' experience in pelvic MRI), using T2WI, contrast-enhanced T1WI, and pelvic examination findings for guidance. Non-enhanced areas which were considered to be cystic, necrotic, and/or hemorrhagic were excluded. If no residual tumor was observed after treatment, the VOI was placed in the area where the lesion had been on the previous MRI. To examine inter-observer reliability, a second radiologist with 15 years'

**Table 1. Summary of MRI acquisition protocols.**

| Sequences | T2WI | T2WI | DWI | CE-T1WI |
|---|---|---|---|---|
| Plane | Sagittal | axial | axial | sagittal |
| TR/TE | 3000/100 | 3000/100 | 5000/90 | 540/15 |
| Field of view(mm) | $240 \times 240$ | $240 \times 240$ | $230 \times 230$ | $240 \times 240$ |
| Matrix size | $480 \times 480$ | $704 \times 704$ | $128 \times 128$ | $384 \times 384$ |
| Slice thickness (mm) | 5 | 4 | 5 | 5 |
| Intersection gap (mm) | 1 | 1 | 1 | 1 |
| Number of excitations | 1 | 1 | 2 | 2 |
| b value (sec/mm$^2$) | | | 1000 | |

*Abbreviations*: *rec* = recurrent, non-rec = non-recurrent.

experience in pelvic MRI independently set VOIs on 96 MRIs (24 MRIs per time point) of 24 cases from those patients who had undergone MRI at each of the four time points during the treatment course.

**2C. Feature extraction.** A total of 107 image features (number of morphology features = 14, histogram features = 18, texture features = 75) were extracted from the VOIs using open-source software, PyRadiomics v3.0 [17,18]. Differences between PyRadiomics and the Image Biomarker Standardization Initiative (IBSI) have been examined and shown to be in agreement [19].

An absolute rescaling method ($0-4 \times 10^{-3}$mm$^2$/sec) was applied to normalize ADC values. Pixel values between the upper and lower limits were resampled into 64 levels.

The change rates in the image features were calculated as follows:

$$Change\ rate = (later\ feature\ value - previous\ feature\ value)/previous\ feature\ value$$

We calculated the change rates from six combinations of two MRI examinations (between the 1st and 2nd, 1st and 3rd, 1st and 4th, 2nd and 3rd, 2nd and 4th, and 3rd and 4th). We refer to the image features from the 1st, 2nd, 3rd, and 4th MRIs and their change rates as 'image parameters'. The protocol for this analysis is shown at protocols io (https://www.protocols.io/view/analysis-protocol-plosone-chx8t7rw).

## 3. Statistics

**3A. Univariate analysis.** Univariate analyses were performed on clinical data including age, FIGO stage (FIGO2008), and histological type using Fisher's exact test and the Mann–Whitney test. Image parameters were compared with the Mann–Whitney test.

**3B. Receiver operating characteristic (ROC) and survival analyses.** In the 57 patients, the recurrence prediction ability of the image parameters was evaluated by ROC analysis. The presence or absence of local and/or distant recurrence within 2 years was set as an outcome. No patients were lost to follow-up. The cut-off values were determined from the ROC curve using the maximum Youden index for image parameters, which shows a high area under the curve (AUC). Kaplan-Meier plots of disease-free survival (DFS) were constructed with the cut-off value, and the log-rank test was used for comparison. We also performed the same ROC analysis only in patients with SqCC, calculated cut-off values, and constructed Kaplan-Meier plots in the same way.

**3C. Inter-observer reliability.** The Dice similarity coefficient (DSC), used to indicate the overlap ratio between the VOIs drawn by two radiologists, was calculated to compare all sets of VOIs. The intraclass correlation coefficient (ICC) was also calculated for the image

parameters with high AUCs in the ROC. DSC scores > 0.70 are generally considered to have good agreement [20]. The ICC values were interpreted as follows: "poor" (ICC < 0.5), "moderate" (0.5–0.75), "good" (0.75–0.9) and "excellent" (ICC > 0.9) [21,22].

## Results

### 1. Eligible MRIs

We included only MRIs taken with the 3T MRI (Ingenia, Philips Medical Systems, Best, Netherland). For 57 patients, 41, 41, 27, and 53 MRIs were available at time points 1, 2, 3, and 4, respectively. (Fig 1). The change rates from six combinations of two MRI examinations were calculated, as shown in Fig 1.

### 2. Patients' clinical characteristics

Twenty-three of 57 patients developed a recurrence within two years, while 34 patients had no recurrence within two years of follow-up. In the 23 patients with recurrence, eight patients had local pelvic recurrence (cervix and lymph nodes) within the irradiation field, and 15 patients developed distant metastasis. Clinical data and the results of the univariate analyses are shown in Table 2. Forty-nine of the 57 patients had SqCC. The results of the analysis in SqCC patients only are shown in the (S1 and S2 Tables, S1 Fig). Recurrence significantly differed by histological type ($p$ = 0.014), with adenocarcinoma recurring more frequently than the other types. In contrast, FIGO stage showed no significant difference between the recurrence and non-recurrence groups.

### 3. ROC analysis

**3A. Summary of ROC analysis.** Receiver Operating Characteristic analysis was performed in the 57 cases; the image parameters with high AUCs are shown in Table 3. The predictive power of the change rates in kurtosis (sharpness of the histogram peaks) between the 1st and 3rd, and 1st and 2nd scans, and the change rate in grey level co-occurrence matrix (glcm)_cluster shade between the 2nd and 3rd scans, were particularly high (AUC = 0.785, 0.759, and 0.750, respectively). Among the image features of the 1st MRI, glcm_Inverse Difference (glcm_Id) had the best prognostic accuracy (AUC = 0.737). Our data set and the AUCs of all image parameters are described in the (S3 and S4 Tables).

**Table 2. Clinical data and results of the univariate analyses.**

|  |  | All (n = 57) | Rec (n = 23) | Non-rec (n = 34) | *p* |
|---|---|---|---|---|---|
| Site of recurrence | local |  | 8 |  |  |
|  | distant |  | 15 |  |  |
| Age (mean ± standard deviation) |  | 61 ± 14.3 | 60.6 ± 13.7 | 61.3 ± 14.7 | 0.839 |
| FIGO | 1B | 15 | 5 | 10 | 0.659 |
|  | 2A | 10 | 5 | 5 |  |
|  | 2B | 14 | 5 | 9 |  |
|  | 3A | 2 | 1 | 1 |  |
|  | 3B | 12 | 5 | 7 |  |
|  | 4A | 4 | 2 | 2 |  |
| Pathological type | SqCC | 49 | 17 | 32 | 0.014 |
|  | Adenocarcinoma | 7 | 6 | 1 |  |
|  | Adenosquamous | 1 | 0 | 1 |  |

**Table 3. Image parameters with high AUCs in all patients.**

| image parameter | status | AUC |
|---|---|---|
| Kurtosis. | change rate 1st–3rd | 0.785 |
| Kurtosis. | change rate 1st–2nd | 0.759 |
| glcm_ClusterShade. | change rate 2nd–3rd | 0.750 |
| glcm_Id. | 1st | 0.737 |
| glcm_Imc2. | change rate 1st–3rd | 0.736 |
| glcm_Idm. | 1st | 0.734 |
| gldm_DependenceVariance. | 1st | 0.734 |
| shape_SurfaceVolumeRatio. | 2nd | 0.733 |
| gldm_DependenceNonUniformityNormalized. | 1st | 0.732 |
| glszm_LargeAreaHighGrayLevelEmphasis. | change rate 1st–3rd | 0.729 |

*Abbreviations*: glcm = grey level co-occurrence matrix; glcm_Id = grey level co-occurrence matrix Inverse Difference; glcm_imc2 = grey level co-occurrence matrix informational measure of correlation; gldm = gray Level dependence matrix; glszm = gray level size zone matrix

**3B. Morphology parameters.** The AUC of shape_VoxelVolume on the 1st MRI (pretreatment tumor volume) as a conventional prognostic factor was 0.614. The change rate in shape_VoxelVolume between the $1^{st}$ and $4^{th}$ MRI, which represents change in tumor volume before and after treatment, showed an AUC of 0.600.

**3C. Histogram parameters.** Among the histogram parameters, the change rates in kurtosis between the $1^{st}$ and $3^{rd}$ MRIs (kurtosis change $1^{st}$–$3^{rd}$) and between the $1^{st}$ and $2^{nd}$ MRIs (kurtosis change $1^{st}$–$2^{nd}$) showed high AUCs (0.785 and 0.759, respectively), namely the highest and the second-highest of all parameters. In contrast, kurtosis change between the $1^{st}$–$2^{nd}$ and $1^{st}$–$3^{rd}$ MRIs were significantly lower in the recurrence group than in the nonrecurrence group ($p$ = 0.018 and 0.020, respectively); that is, kurtosis showed a greater decrease between the $1^{st}$ and the $2^{nd}$ or $3^{rd}$ MRI in the recurrence group (Table 4).

**3D. Texture parameters.** Among the texture parameters, the change rate in glcm_ClusterShade between the $2^{nd}$ and $3^{rd}$ MRIs (glcm_ClusterShade change $2^{nd}$–$3^{rd}$) showed the highest AUC (0.750), and was third highest of all parameters. The glcm_ClusterShade change $2^{nd}$–$3^{rd}$ was significantly higher in the recurrence group than in the nonrecurrence group ($p$ = 0.036) (Table 4).

## 4. Survival analysis

We determined the cut-off values for the three parameters with high AUCs (kurtosis change $1^{st}$–$3^{rd}$, kurtosis change $1^{st}$–$2^{nd}$, and glcm_ClusterShade change $2^{nd}$–$3^{rd}$) using the maximum Youden index. Kaplan-Meier plots were created with the cut-off values (Fig 3). Each parameter divided the patients into those with better or worse prognoses ($p$ = 0.032, 0.0087, and 0.008, respectively). Box-and-whisker plots of kurtosis in the $1^{st}$ to $4^{th}$ MRI in the recurrent and non-recurrent groups are shown in Fig 4.

**Table 4. Mean and standard deviation (SD) in the change rate of kurtosis and glcm_ClusterShade in the recurrent and non-recurrent groups.**

| | Non-recurrent | | recurrent | | p-value |
|---|---|---|---|---|---|
| | mean | SD | mean | SD | |
| kurtosis change 1st– 2nd | −0.036 | 0.310 | −0.317 | 0.153 | 0.018 |
| kurtosis change 1st– 3rd | −0.126 | 0.654 | −0.452 | 0.171 | 0.020 |
| glcm_ClusterShade change 2nd– 3rd | −18.682 | 40.515 | −1.143 | 0.498 | 0.036 |

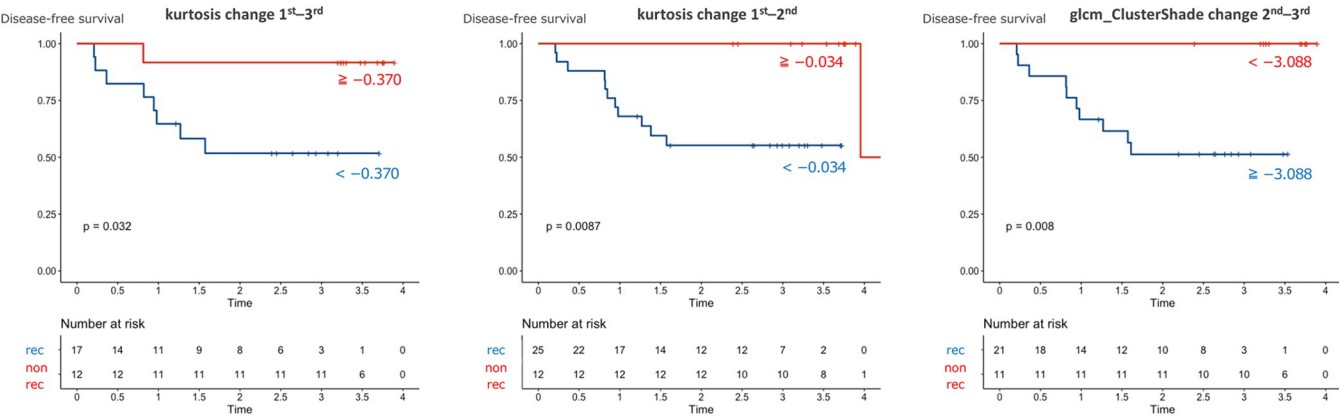

**Fig 3. Kaplan-Meier plots for all patients.** Log-rank *p* values were calculated. X axis: Time (years), Y axis: Disease-free survival (probability). Number of patients at risk is shown below. glcm = grey level co-occurrence matrix.

## 5. Inter-observer reliability

The mean DSC scores for the VOIs of each imaging time point are shown in Table 5A. ICC scores of kurtosis and glcm_ClusterShade at each imaging time point and the three image parameters are shown in Table 5B. The mean DSC scores of the 1st and 2nd MRI were good (0.852 and 0.763), while the DSC scores of the 3rd and 4th MRI were poor (0.430 and 0.438). Kurtosis of the 1st and 2nd MRI and the kurtosis change $1^{st} - 2^{nd}$ had moderate-to-good reliability (ICC = 0.783, 0.729, and 0.765, respectively), while kurtosis of the $3^{rd}$ MRI and the kurtosis change $1^{st} - 3^{rd}$ had poor reliability (ICC = 0.222 and 0.408). The value of Glcm_ClusterShade of 2nd and 3rd MRIs and the glcm_ClusterShade change $2^{nd} - 3^{rd}$ had poor reliability (ICC $\leq$ 0.001, 0.106, and < 0.001, respectively).

## Discussion

Image features of ADC values and their change rates during the disease course and treatment may predict prognosis with high accuracy in CC patients. In this study, the predictive power of the kurtosis changes $1^{st} - 3^{rd}$ and $1^{st} - 2^{nd}$, and the glcm_ClusterShade change $2^{nd} - 3^{rd}$ were particularly high (AUC $\geq$ 0.75). The prognostic prediction abilities of these change rates were higher than the parameters obtained from pretreatment or posttreatment MRI only. These three change rates also showed high predictive accuracy in the analysis of SqCC patients. The kurtosis change $1^{st} - 2^{nd}$ was considered the most reliable because the kurtosis of the $1^{st}$ and $2^{nd}$ MRIs and the kurtosis changes between them showed high ICC scores.

Kurtosis reflects the degree of sharpness of the histogram and the combined weight of the histogram tails relative to the rest of the distribution. The kurtosis increases as the peak becomes higher and the tails become heavier, while it decreases as the peak becomes flatter and the tails become lighter. A high peak suggests homogeneity, and a flat peak suggests variability or heterogeneity [23]. Lower tumor ADC kurtosis may indicate the heterogeneity of tumor ADC values, although histogram tails or outliers somehow affect kurtosis [5,24]. Ciolina et al. reported that lower tumor ADC kurtosis in pretreatment MRI might reflect more disorganized tumor tissues with 'nuclear enlargement and an ineffective vascular net', which reduced the efficacy of chemotherapy in CC [5]. Shaghaghi et al. reported that lower tumor kurtosis represents an increase in tissue heterogeneity and scattered necrosis within the tumor in HCC treated with transcatheter arterial chemoembolization [25]. During treatment, a decrease in tumor ADC probably reflects therapy-induced heterogeneous changes, which may

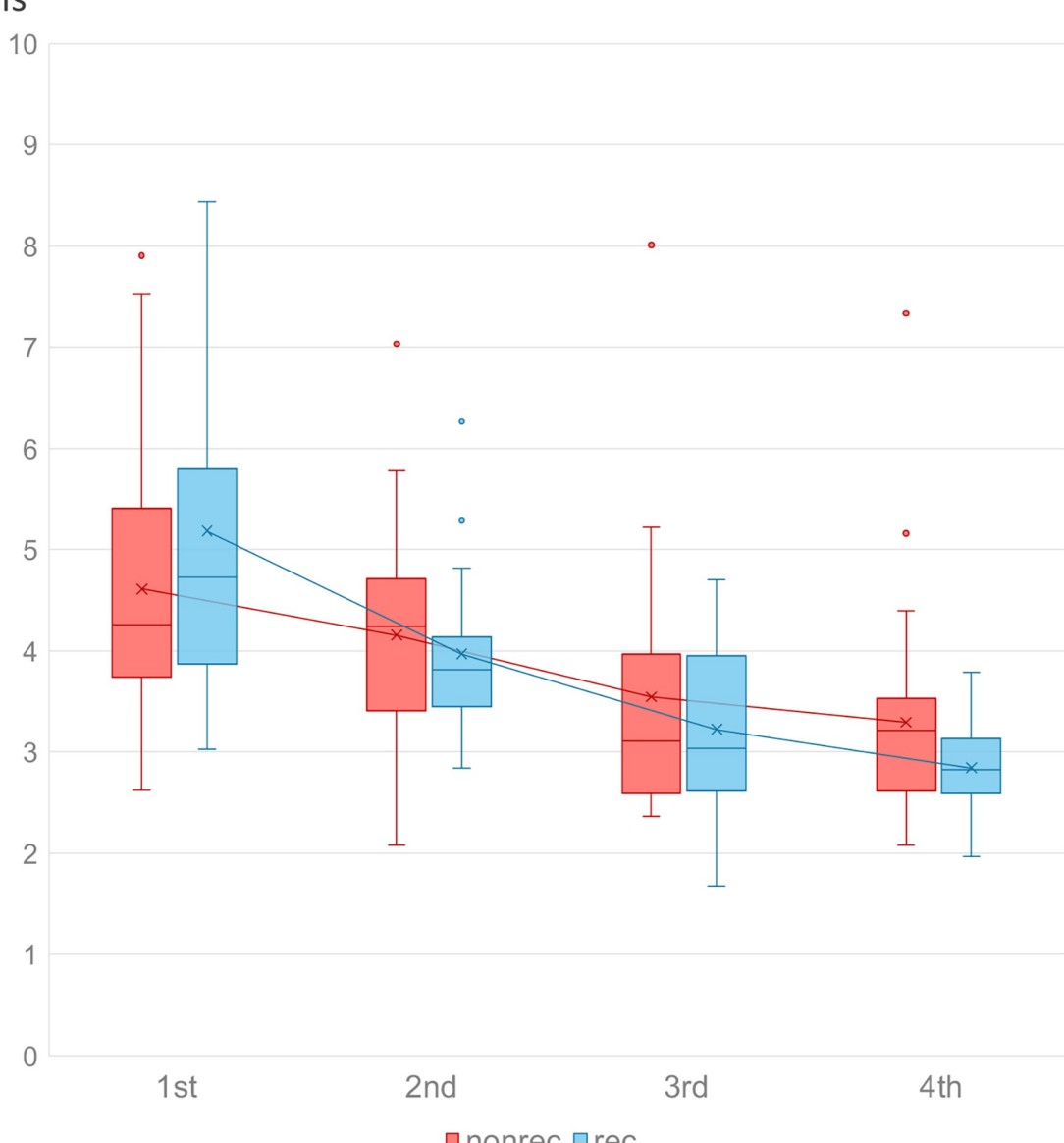

**Fig 4. Box-and-whisker plots of ADC kurtosis.** Box-and-whisker plots show trends in ADC kurtosis during the clinical course in the recurrent and non-recurrent groups. The mean value of kurtosis is indicated by the × symbol. Tumor ADC kurtosis tended to decrease towards the end of treatment as an overall trend in both groups. The decrease in kurtosis from the first to the third MRI was greater in the recurrent than in the non-recurrent group.

include scattered necrosis, destroyed tumor vasculature, and nonuniform tumor cell morphologies.

In our study, ADC kurtosis tended to decrease more in the recurrent group than in the nonrecurrent group, particularly between the 1st and 2nd or 3rd MRI (Fig 4). Increased intratumoral necrotic and hypoxic tissue cause resistance to radiation therapy [26]. Therefore, a greater decrease in tumor ADC kurtosis between the 1st and 2nd /3rd MRI, representing extensive heterogenic and scattered necrotic changes within the tumor early during CCRT, could be a predictor of poor prognosis for CC treated with CCRT. Examples of ADC histograms of recurrent and non-recurrent cases are shown in Fig 5. Contrary to the findings with kurtosis,

**Table 5.  a. Dice similarity coefficient (DSC) scores (average and standard deviation) in the 1st–4th time points.** b. Intraclass correlation coefficients (ICC) values of the ADC kurtosis and the glcm_ClusterShade in the 1st–4th time points.

| | 1st | | 2nd | | 3rd | 4th | |
|---|---|---|---|---|---|---|---|
| mean | 0.852 | | 0.763 | | 0.43 | 0.438 | |
| SD | 0.073 | | 0.164 | | 0.291 | 0.242 | |
| | 1st | 2nd | 3rd | 4th | change rate 1st–2nd | change rate 1st–3rd | change rate 2nd–3rd |
| kurtosis | 0.783 | 0.729 | 0.222 | < 0.001 | 0.765 | 0.408 | – |
| glcm_ClusterShade | 0.714 | < 0.001 | 0.106 | 0.580 | – | – | < 0.001 |

*Abbreviations*: glcm = grey level co-occurrence matrix.

glcm_ClusterShade, a measure of the skewness and uniformity of the GLCM, tended to decrease less markedly between the 2nd and 3rd MRI in the recurrent group. A higher glcm_ClusterShade implies greater asymmetry about the mean [27]. A higher glcm_Cluster-Shade may also reflect the heterogeneity of the tumor. Therefore, a small decrease in glcm_ClusterShade might predict a poor prognosis.

The ICCs of kurtosis tended to decline as treatment progressed and were significantly associated with the shrinkage or disappearance of the tumor. When the tumor had disappeared, the VOI setting where the lesion was located at the previous MRI is subjective. A past study also reported that smaller tumor volumes contributed to poor inter-observer reliability of ADC [28]. The effects associated with gas in the intestine or in areas of necrosis will likely be more significant in small lesions than in large lesions. This may be improved by the administration of scopolamine butylbromide before MRI acquisition and efforts not to include air during segmentation. Another cause is that the therapeutic effects obscure the boundary between the residual tumor and surrounding normal tissue, making tumor segmentation difficult [29,30]. The DSC scores were also lower at the 3rd and 4t than at the 1st and 2nd MRIs. Because of the high ICC, the kurtosis change 1st–2nd is an excellent parameter with high predictive accuracy and reliability.

A few studies have predicted long-term prognosis using intra-treatment MRI, but the results differed from our present results [8,31,32]. Two of these studies reported that changes in mean tumor ADC during treatment and energy on ADC maps of intra-treatment MRI showed high predictive power [8,31]. In our study, mean intra-treatment ADC, energy, and their change rates did not significantly differ between the recurrent and nonrecurrent groups (AUC < 0.72, supporting information). In addition, the change rate in kurtosis was not useful in predicting prognosis in a past study [32]. These discrepancies may be due to the following factors: (1) differences in MR protocols and parameters, (2) different sample size, and (3) differences in the VOI placement method with or without non-enhanced areas within the tumor. A previous study that reported low predictability of kurtosis determined VOIs including cystic and necrotic areas within the tumor [32], although it is common to exclude non-enhanced areas to avoid outliers in imaging analyses for CC [8,33,34]. Kurtosis can be susceptible to error because it is affected not only by the degree of sharpness but also by the tails of the histogram. Air and hemorrhage in necrotic tissue may hinder the accurate calculation of tumor ADC kurtosis.

Our study has some limitations. First, we had a relatively small study population, and recurrent cases included both local and distant recurrence. Nevertheless, our study has the largest number of patients among papers about long-term prognosis prediction using change rates in

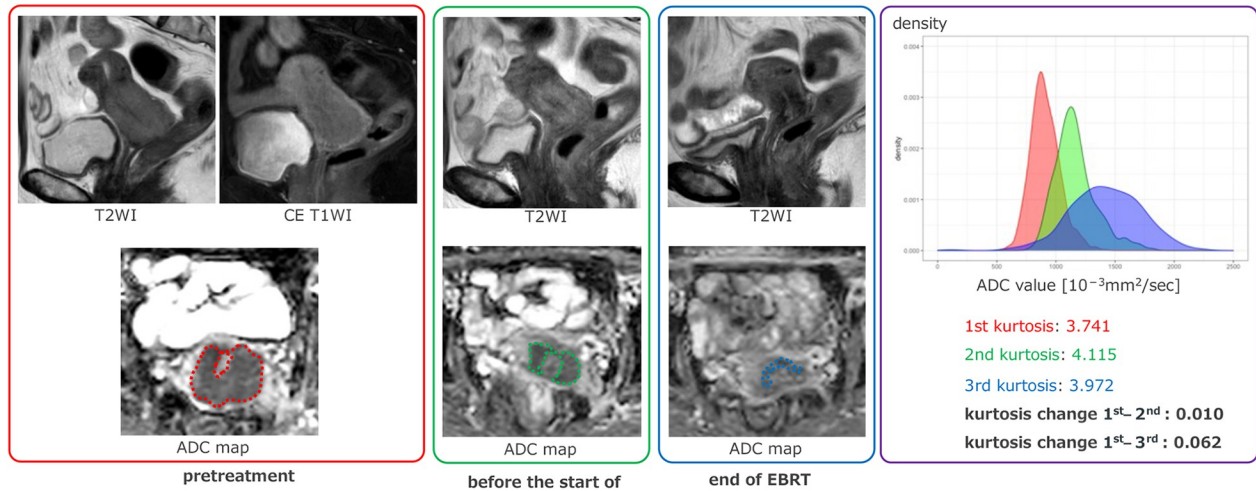

**Fig 5. a. Recurrent case.** A 66-year-old patient with FIGO stage IIB squamous cell carcinoma experienced recurrence with distance lymph node metastasis in a year. An intra-tumoral T2WI hypointense area (arrow) with poor enhancement on T1WI was presumed to be necrosis. The necrotic area expanded on the following 2nd and 3rd scans. Although segmentation was performed to exclude necrosis, the decrease in kurtosis is conspicuous in the 2nd and 3rd scans. The sharpness of the histogram also decreases in the 2nd and 3rd scans, indicating decreasing kurtosis. **b. Non-recurrent case.** A 70-year-old patient with FIGO stage IIB squamous cell carcinoma had been followed up for more than two years without sign of recurrence. No apparent necrosis was observed in the tumor in the 2nd and 3rd scans. Kurtosis did not decrease, and there was no obvious decrease in histogram sharpness.

the ADC histogram and texture features during treatment for CC. Second, not all patients had been able to undergo MRI four times during the study, and the number of available MRIs differed depending on the time point. In particular, the number of 3rd MRIs was relatively small. Third, we could not pathologically confirm what tumor ADC kurtosis reflected. Further study is needed to examine what the image parameters reflect pathologically in surgical specimens.

In conclusion, image features and their change rates in ADC during the treatment course were useful for predicting the prognosis after definitive CCRT for CC. They reflected the early response of the tumor tissue to CCRT; therefore, image parameters using intra-treatment MRI showed high predictive accuracy. The kurtosis change 1st–2nd was the most reliable and has the advantage of predicting prognosis early during treatment before the initiation of IGBT.

Predicting prognosis early during treatment enables the treating physicians to adjust the radiation and chemotherapy dose and to provide a more personalized treatment.

## Supporting information

**S1 Fig. Kaplan-Meier plots for SqCC patients.** Log-rank *p* values were calculated. *Abbreviations*: glcm = grey level co-occurrence matrix.
(DOCX)

**S1 Table. Clinical data and the results of univariate analyses of SqCC patients only.**
(DOCX)

**S2 Table. The image parameters with high AUCs in the 49 SqCC patients.**
(DOCX)

**S3 Table. Data set of features.**
(XLSX)

**S4 Table. The AUCs of all image parameters.**
(DOCX)

**S1 File.**
(DOCX)

## Author Contributions

**Conceptualization:** Akiyo Takada, Hajime Yokota.

**Data curation:** Akiyo Takada, Miho Watanabe Nemoto, Takuro Horikoshi, Koji Matsumoto, Yuji Habu, Hirokazu Usui.

**Formal analysis:** Akiyo Takada.

**Investigation:** Akiyo Takada.

**Methodology:** Akiyo Takada, Hajime Yokota, Koji Matsumoto.

**Project administration:** Akiyo Takada.

**Writing – original draft:** Akiyo Takada.

**Writing – review & editing:** Akiyo Takada, Hajime Yokota, Katsuhiro Nasu, Makio Shozu, Takashi Uno.

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
