## [Decision Letter · Decision Letter 0]

20 Sep 2022

PONE-D-22-13961Prognosis prediction of uterine cervical cancer using changes in the histogram and texture features of apparent diffusion coefficient during definitive chemoradiotherapyPLOS ONE

Dear Dr. Takada,

Thank you for submitting your manuscript to PLOS ONE. After careful consideration, we feel that it has merit but does not fully meet PLOS ONE’s publication criteria as it currently stands. Therefore, we invite you to submit a revised version of the manuscript that addresses the points raised during the review process. Please submit your revised manuscript by Nov 04 2022 11:59PM. If you will need more time than this to complete your revisions, please reply to this message or contact the journal office at plosone@plos.org. Please include the following items when submitting your revised manuscript:A rebuttal letter that responds to each point raised by the academic editor and reviewer(s). You should upload this letter as a separate file labeled 'Response to Reviewers'.A marked-up copy of your manuscript that highlights changes made to the original version. You should upload this as a separate file labeled 'Revised Manuscript with Track Changes'.An unmarked version of your revised paper without tracked changes. You should upload this as a separate file labeled 'Manuscript'.

We look forward to receiving your revised manuscript.

Kind regards,

Ngie Min Ung

Academic Editor

PLOS ONE

2. Thank you for including your ethics statement:  "The ethical committee of our institute approved this study and informed consent was obtained from each patient". 

For studies reporting research involving human participants, PLOS ONE requires authors to confirm that this specific study was reviewed and approved by an institutional review board (ethics committee) before the study began. Please provide the specific name of the ethics committee/IRB that approved your study, or explain why you did not seek approval in this case.

Reviewers' comments:

Reviewer's Responses to Questions

**Comments to the Author**

1. Is the manuscript technically sound, and do the data support the conclusions?

Reviewer #1: Partly

Reviewer #2: Partly

2. Has the statistical analysis been performed appropriately and rigorously? 

Reviewer #1: Yes

Reviewer #2: Yes

3. Have the authors made all data underlying the findings in their manuscript fully available?

Reviewer #1: Yes

Reviewer #2: Yes

4. Is the manuscript presented in an intelligible fashion and written in standard English?

Reviewer #1: Yes

Reviewer #2: No

5. Review Comments to the Author

Reviewer #1: You work is a subject of great interest .

A lot of work has already been done for Response assessment using texture features in Cervix cancer. This work of your though extensive is unfortunately not unique.

I have major concerns in the methods section : Why are tumor markers listed ? Are they specific of SCC Cervix cancer ? Why was CT done - to look for nodes ? If yes what was the criteria ? Why and how was PET-CT used ? Please specify ? Was it still biopsied after PET- What was SUV cut off ? What was the standard of truth - HPE or Clinical assessment ? This has to be one of the major flaws in this manuscript which wasn't addressed .

What was the significance of a "2 year recurrence " follow up ? This has to be elaborated more in all parts of the manuscript !

Another major flaw - Mentioned in manuscript that areas of necrosis were omitted at time of segmentation- Past literature shows necrosis was infact more inclusive of overall prognosis .

Reviewer #2: Overall comments:

The manuscripts provide valuable information which would support the outcomes and endpoints of cervical cancer treatment in radiotherapy. However, the manuscript lacks detailing radiotherapy procedure in which this information would be directly related to the data collected in this study.

1) The English language could be proofread - This does not affect the clarity of contents but could nevertheless be improved.

a. There are a few instances where the spelling is following both the American and British styles (tumor vs tumour, gynecology vs gynaecology, hemorrhage vs haemorrhage).

b. There are few sentences started with abbreviations, in which this normally should be avoided in a formal and scientific writing. I am giving two examples from the manuscript;1) DSCs of the 1st and 2nd MRI were good (0.852 and 0.763), while DSCs of the 3rd and 4th MRI were poor... 2) Glcm_ClusterShade of 2nd and 3rd MRIs and the glcm_ClusterShade change 2nd –3rd had poor reliability...

2) I find the writing of references does not follow the correct format. I suggest the author to review and check the correct format for reference writing according to journal requirements. In addition to this, each reference in the sentences does not link to the reference list. This arises difficulties to the reviewer to cross-check the reference. It would be good if this was included in the manuscript. Author can use the References Tab in the words document for cross-reference link.

3) This manuscript needs to be re-written in more rigorous and scientific manner. Although the study provides meaningful data and information, I believe the clarification on data presented in this manuscript is being blinded in the use of non-scientific manner writing.

Specific Comments:

Please refer to the comment boxes inserted in the uploaded reviewer's attachment.

6. PLOS authors have the option to publish the peer review history of their article (what does this mean?). If published, this will include your full peer review and any attached files.

Reviewer #1: **Yes: **ANKUSH JAJODIA

Reviewer #2: No

---

## [Author Response · Author response to Decision Letter 0]

20 Nov 2022

I have reassured that my manuscript met PLOS ONE's style requirements.

2. Thank you for including your ethics statement: "The ethical committee of our institute approved this study and informed consent was obtained from each patient". 

For studies reporting research involving human participants, PLOS ONE requires authors to confirm that this specific study was reviewed and approved by an institutional review board (ethics committee) before the study began. Please provide the specific name of the ethics committee/IRB that approved your study, or explain why you did not seek approval in this case.

I have added statements about approval of ethical committee including approval number.

I have added statements about participant concent. We got written informed consent from all patients.

4. In your Data Availability statement, you have not specified where the minimal data set underlying the results described in your manuscript can be found. PLOS defines a study's minimal data set as the underlying data used to reach the conclusions drawn in the manuscript and any additional data required to replicate the reported study findings in their entirety. All PLOS journals require that the minimal data set be made fully available. For more information about our data policy, please see http://journals.plos.org/plosone/s/data-availability

We have uploaded our protocols to protocols io, and added new Supporting Information file including our whole dataset.

---

## [Decision Letter · Decision Letter 1]

22 Feb 2023

Prognosis prediction of uterine cervical cancer using changes in the histogram and texture features of apparent diffusion coefficient during definitive chemoradiotherapy

PONE-D-22-13961R1

Dear Dr. Takada,

We’re pleased to inform you that your manuscript has been judged scientifically suitable for publication and will be formally accepted for publication once it meets all outstanding technical requirements.

Kind regards,

Ngie Min Ung

Academic Editor

PLOS ONE

Additional Editor Comments (optional):

Reviewers' comments:

Reviewer's Responses to Questions

**Comments to the Author**

1. If the authors have adequately addressed your comments raised in a previous round of review and you feel that this manuscript is now acceptable for publication, you may indicate that here to bypass the “Comments to the Author” section, enter your conflict of interest statement in the “Confidential to Editor” section, and submit your "Accept" recommendation.

Reviewer #2: All comments have been addressed

2. Is the manuscript technically sound, and do the data support the conclusions?

Reviewer #2: Yes

3. Has the statistical analysis been performed appropriately and rigorously? 

Reviewer #2: Yes

4. Have the authors made all data underlying the findings in their manuscript fully available?

Reviewer #2: Yes

5. Is the manuscript presented in an intelligible fashion and written in standard English?

Reviewer #2: Yes

6. Review Comments to the Author

Reviewer #2: Very minor changes need to be mad in the methodology section. The author stated the external beam radiotherapy delivered with Ir-192. Please refer to the comment box in the attached file.

7. PLOS authors have the option to publish the peer review history of their article (what does this mean?). If published, this will include your full peer review and any attached files.

Reviewer #2: No

---

## [Editor Report · Acceptance letter]

21 Mar 2023

PONE-D-22-13961R1 

Prognosis prediction of uterine cervical cancer using changes in the histogram and texture features of apparent diffusion coefficient during definitive chemoradiotherapy 

Dear Dr. Takada:

I'm pleased to inform you that your manuscript has been deemed suitable for publication in PLOS ONE. Congratulations! Your manuscript is now with our production department. 

Kind regards, 

on behalf of

Dr. Ngie Min Ung 

Academic Editor

PLOS ONE